# Hydrological Modelling in Data Sparse Environment: Inverse Modelling of a Historical Flood Event

**András Bárdossy** **, Faizan Anwar \*** and **Jochen Seidel**

Institute for Modelling Hydraulic and Environmental Systems, University of Stuttgart,
D-70569 Stuttgart, Germany; andras.bardossy@iws.uni-stuttgart.de (A.B.);
jochen.seidel@iws.uni-stuttgart.de (J.S.)
**\*** Correspondence: faizan.anwar@iws.uni-stuttgart.de

**Abstract:** We dealt with a rather frequent and difficult situation while modelling extreme floods, namely, model output uncertainty in data sparse regions. A historical extreme flood event was chosen to illustrate the challenges involved. Our aim was to understand what the causes might have been and specifically to show how input and model parameter uncertainties affect the output. For this purpose, a conceptual model was calibrated and validated with recent data rich time period. Resulting model parameters were used to model the historical event which subsequently resulted in a rather poor hydrograph. Due to the bad model performance, a spatial simulation technique was used to invert the model for precipitation. Constraints, such as taking the precipitation values at historical observation locations in to account, with correct spatial structures and following the observed regional distributions were used to generate realistic precipitation fields. Results showed that the inverted precipitation improved the performance significantly even when using many different model parameters. We conclude that while modelling in data sparse conditions both model input and parameter uncertainties have to be dealt with simultaneously to obtain meaningful results.

**Keywords:** inverse modelling; data uncertainty; parameter uncertainty; data scarcity

---

## 1. Introduction

Floods are rare extreme events with high impact on society and human life. The knowledge of possible flood occurrences and magnitudes is very important for the design of flood protection measures. In most cases, extreme value statistics is used to estimate probabilities of excedance of such events. This requires long discharge time series and the knowledge of the corresponding flood peaks. Therefore, information on historical flood events offers the potential to improve flood risk assessment, but their proper treatment can be difficult Benito et al. [1]. Generally, flood peaks are reconstructed from past records based on flood marks and the investigation of the hydraulic conditions Glaser et al. [2]. For Germany, systematic studies using this approach have been presented by e.g., Sudhaus et al. [3] and Herget et al. [4]. For the period of time from which early instrumental hydro-meteorological data are available, it is also possible to reconstruct the hydro-meteorological conditions that caused extreme floods (e.g., Bürger et al. [5], Seidel et al. [6], Bomers et al. [7]). This approach is interesting in so far as recently more attention was driven to understand flood generating mechanisms and to quantify their effect on the probability of excedance in Fischer and Schumann [8].

In this contribution, we investigated a single historical flood event in the Neckar Catchment (SW-Germany) for which the flood hydrograph was reconstructed in a previous study by Sudhaus et al. [3] by using historical profiles and modelled by Seidel et al. [6]. Their aim was to understand the flood generating mechanism and to model the rainfall-runoff by finding precipitation analogues in the recent data before and during the event. We use an inverse modelling approach to

reconstruct the possible flood triggering precipitation, mainly to account for the spatial variability of precipitation which is not captured by sparse early instrumental observations. Inverting models to obtain plausible model parameters or input precipitation is not unheard of in hydrology and has become a more frequent field of research, e.g., Kirchner [9], Herrnegger et al. [10], Wright et al. [11], Boudhraâ et al. [12], Wright et al. [13], Grundmann et al. [14], for gaining a better understanding of the relationships among observations and model outputs/parameters.

Specifically, here, we intended to,

- check the flood-peak-magnitude plausibility from the hydrological perspective and
- understand and quantify the flood generating processes and
- differentiate between model output uncertainty caused by uncertain model parameters and inputs, especially in a data sparse environment.

For this purpose, we used a conceptual hydrological model to describe the selected event. Hydrological modelling of historical events is difficult, comparatively, as only a few meteorological observations are available and have inferior quality because they were not measured using state of the art devices and practices Brázdil et al. [15]. We used a model calibrated under present data rich conditions for modelling the selected historical discharge extreme of the year 1882 in the upper Neckar catchment in South-West Germany. As precipitation observations for the selected event were sparse, an inverse modelling approach was used. With the help of the reconstructed discharge and the precipitation measurements, we tried to find plausible precipitation patterns for the selected event which match the historical observations. Thus, we searched for possible precipitation fields that confirmed the available observations of precipitation and were able to explain the discharges prior to, during and after the event. There are modelled results of two distinct time periods. The "present day data" refers to the time data rich period between 1961–2015, while the "historical data", refers to the data sparse year of 1882.

This paper is organized as follows: The investigation area (Section 2.1) and data (Section 2.2) follow the introduction. Overview of the hydrological model and its calibration are given in Sections 2.4 and 2.5. Model inversion methodology is given in Section 2.6. Results of the calibration and inversion are shown and discussed in Section 3. Finally, conclusions for the reconstructed event are drawn and implications for the general practice of hydrological modelling under data sparse conditions are discussed in Section 4.

## 2. Material and Methods

### 2.1. Study Area

The study area is the upper Neckar catchment in the Federal State of Baden-Württemberg in South-West Germany (Figure 1) up to the City of Tübingen. Unfortunately, the historical gauging station in Tübingen (where a hydrograph for the year 1882 is available from Sudhaus et al. [3] and Seidel et al. [6]) was decommissioned long before the start of the present data rich time period. Therefore, the next operational downstream gauging station at Kirchentellinsfurt, 8.5 km downstream, was chosen for this study. The catchment size at Kirchentellinsfurt is 2320 km$^2$ whereas the catchment area of the former gauging station at Tübingen comprised of 1900 km$^2$. The difference in area was accounted for by simply multiplying the historical discharges by a factor of 1.22 which is the ratio of the catchment areas. Our reason for choosing such a crude approach is that the additional area i.e., about 22%, was not so different than the upstream. We assumed that it will not affect the results significantly.

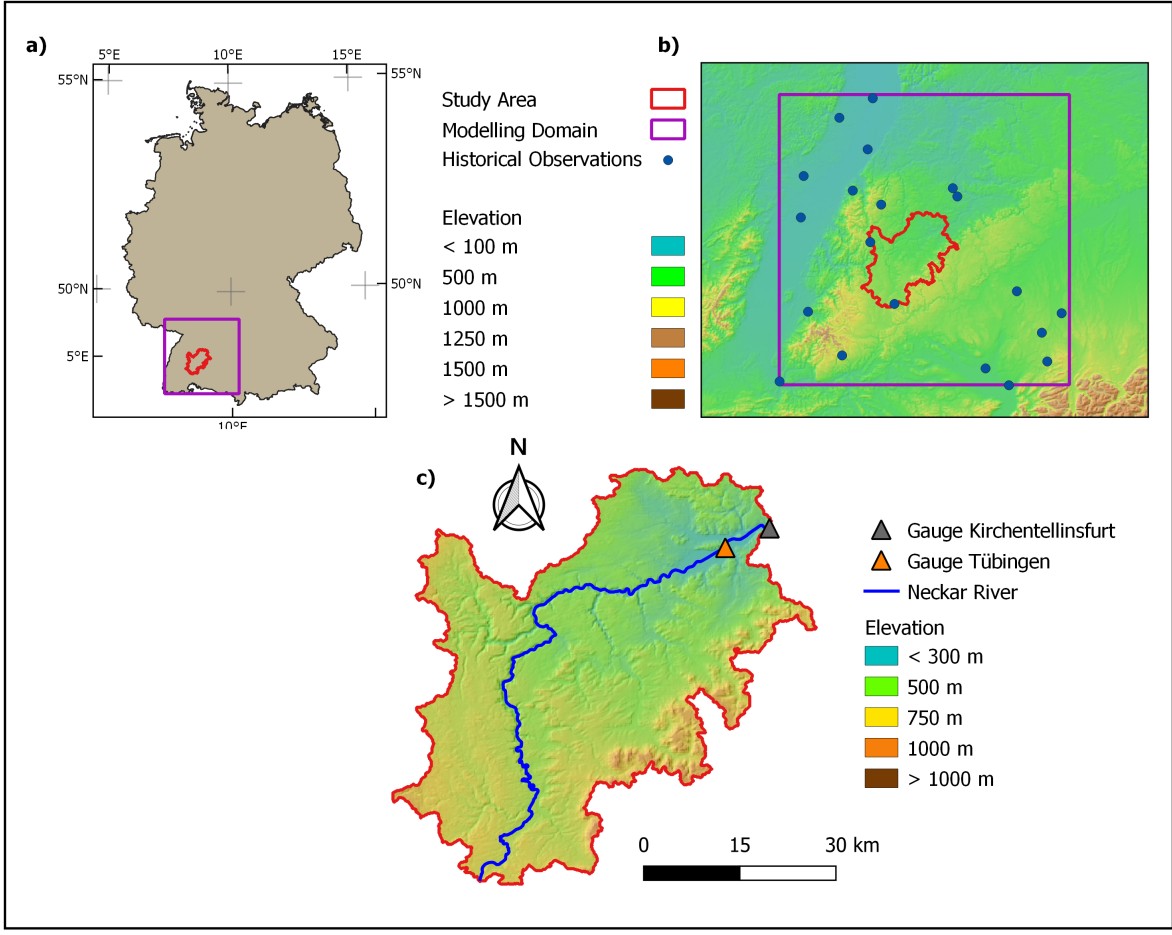

**Figure 1.** Maps showing (**a**) the location of the study area in Germany, (**b**) the modelling domain with the sites of the historical precipitation gauges and (**c**) the catchment area of the gauge at Kirchentellinsfurt.

## 2.2. Data

The daily present day precipitation and temperature data from 1961 to 2000 were obtained from the German Weather Service (DWD) [16]. Daily discharge data are provided by the State Institute for the Environment Baden-Württemberg (LUBW) [17]. Precipitation and temperature were interpolated using Ordinary Kriging (OK) as described in Wackernagel [18]. Potential evapotranspiration (PET) was estimated according to Hargreaves and Samani [19] using the interpolated temperatures.

For the historical flood modelling the input data were collected from various historical sources described in Seidel et al. [6]. As a first step, the daily precipitation observations from 20 stations in the wider catchment area as well as the temperature data were interpolated using the inverse distance weighting (IDW) method described in Shepard [20] and PET was estimated from the interpolated temperature afterwards. OK was not chosen this time as there are very few stations for the year 1882 in/around the study area and this would have resulted in a very unstable variogram which, consequently, would have made the initial results even worse. Using a typical variogram for this month would also result in a very smooth field.

## 2.3. Hydrometeorological Conditions before and during the Flood Event

A detailed analysis of the 1882 flood event in the Neckar Catchment was presented by Seidel et al. [6]. Based on the available meteorological observations at that time, they were able to reconstruct the hydrometeorological conditions that caused this flood. Prior to the event itself, heavy snowfall occurred in large parts of the Neckar catchment area. On December 25, a westerly

circulation pattern brought warm air masses to Central Europe resulting in an air temperature increase of up to 10 °C within 24 h from 25 to 26 December. In addition, persistent rainfall continued until December 27, causing the snow that had previously fallen to melt rapidly. This strong temperature increase in combination with rainfall led to large scale snow melting even at higher altitudes of the Black Forest mountains. Information from contemporary meteorological and newspaper reports state that the snow which had fallen from 22 December onwards, had completely melted by the evening of the 26 December. Hence, the combination of snowmelt and rain led to severe flooding in the tributaries of the Upper Rhine Valley and the Rhine River itself [6].

### 2.4. Hydrological Model Setup

A fully distributed spatial modelling approach using the HBV model [21] was chosen to model the hydrological processes, i.e., each cell in the catchment was modelled separately with its own input data. It goes without saying, the so called physically-based models are not used due to data sparsity. The modelling setup was based on Das et al. [22]. A slight change was taken for the snow melt module of the HBV to account for the circumstance that the historical flood in 1882 was a rain-on-snow event [6]. Warm precipitation induces snow melt i.e., if snow exists and it rains on top then the warm rain can melt snow in addition to the melt caused by warm temperatures (a minor variation of Francés et al. [23]). Such a simple approach sufficed here as snow accumulation in the study region is not so large and snow melt is caused by warmer air temperatures predominantly. A full description/implementation of the HBV model is not provided, as being a fairly known and documented model. Readers interested in our implementation (C++ and Cython) are referred to Anwar and Bárdossy [24]. The modified snow melt routine is given by Equations (1)–(3).

$$ME_i = max(0.0, (CM_{TE} + (CM_{PR} \times PR_i)) \times (TE_i - TT)) \tag{1}$$

$$SN_i = \begin{cases} SN_{i-1} + PR_i & \text{if } TE_i <= TT, \\ max(0.0, SN_{i-1} - ME_i) & \text{else.} \end{cases} \tag{2}$$

$$LP_i = \begin{cases} 0.0 & \text{if } TE_i <= TT, \\ PR_i + min(SN_{i-1}, ME_i) & \text{else.} \end{cases} \tag{3}$$

where the subscript $i$ is the index of a given day, $CM_{TE}$ is the snow melt due to increase in temperature in mm/°C/day, $PR_i$ is the precipitation in mm/day, $CM_{PR}$ is the snow melt due to falling liquid precipitation in mm/°C/day/mm of $PR_i$, $TE_i$ is the temperature in °C, $TT$ is the threshold temperature below which the precipitation falls as snow, $ME_i$ is the possible snow melt in mm, $SN_i$ is the total accumulated snow in mm, $LP_i$ is the liquid precipitation in mm that might come from snow melt or precipitation or both.

The model was set up for the catchment and calibrated and validated using present time meteorological and discharge data. The calibration time period was from 1991 to 2000 (10 years) and the validation from 1961 to 1990 (30 years). The Nash-Sutcliffe efficiency (NSE) [25] was used as the model performance criterion.

### 2.5. Hydrological Model Calibration

Two different calibration methods were used, namely Differential Evolution (DE) [26] and Robust Parameter Estimation (ROPE) [27]. The reason for this was to test two different approaches used in modelling. DE gives only one parameter vector (if there is one) while ROPE produces many, such that all have almost the same performance but vectors can be different from each other. There are advantages and disadvantages to using single- and multi-parameter vectors for evaluating model performance. The interested reader is referred to Cullmann et al. [28]. The main reason to use DE was to have a single model parameter vector that served as the basis for the inverse modelling of

precipitation of the historical flood. ROPE produces more robust results, by having some uncertainty bounds, at the cost of a larger computational time.

To see from the historical year's perspective, the model was also calibrated for the year of 1882 using DE and ROPE. The resulting parameter vectors were used to calculate the performance of the present day calibration and validation periods.

*2.6. Historical Precipitation Simulation*

We focused mainly on simulating precipitation fields. The other variables, temperature and PET, were interpolated because they show comparatively much smoother variability in space and time. The aim was to find daily precipitation fields in high spatial resolution for the flood relevant time period such that the series:

1. match the observed precipitation measured at the observation locations;
2. have values which match the distribution of the observed values of the same day;
3. have spatial variability which does not contradict the observations and have the variogram of a typical day of the season;
4. and using them as input for the hydrological model, the resulting calculated discharge matches the observed discharge well.

To take the initial conditions into account, the hydrological model was run for the whole year of 1882 while the performance was evaluated for November and December only. Precipitation fields for this period were altered according to the aforementioned constraints. Fields for days with low precipitation were not simulated, as their influence on the discharge was negligible. For these days the interpolated precipitation (using IDW) was used instead. The days $t_1 < \ldots < t_m$ with high precipitation amounts measured at any of the observation locations were selected.

Let $x_1, \ldots, x_n$ be the observation locations where precipitation was measured for the historical event. Appropriate precipitation fields are obtained using a stochastic simulation procedure conditioned on the precipitation and discharge observations. Precipitation fields corresponding to each selected day $t_j$ were altered individually. This was repeated for each selected day in a random sequence several times until the modelled discharge was close enough to the observed.

For each of the selected days $t_j$, we assumed that the spatial distribution function of precipitation amounts was the same as the distribution function of the observations $Z(x_i, t_j)$:

$$F_{t_j}(z) = \frac{\mu\left(Z(x, t_j) < z \quad x \in D\right)}{\mu(D)} \approx \frac{\#\left\{i,\ Z(x_i, t_j) < z\right\}}{n} \tag{4}$$

here $\mu$ is the area, $D$ is the catchment under consideration and # is the number of values that satisfy the condition inside the curly braces.

This distribution was estimated by fitting a non-parametric distribution [29] to the observed $Z(x_i, t_j), i = 1, \ldots, n$. Due to the skewness of the observed precipitation amounts the fitting of the distribution was performed after a log transformation. In order to avoid unreasonably high amounts, the distribution was limited such that no values exceeding the maximum observed precipitation by more than 10% were allowed. The resulting distribution was:

$$F_{t_j}(z) = 1 \quad \text{if } z > 1.1 max\left(Z(x_i, t_j), i = 1, \ldots, n\right) \tag{5}$$

This assumption ensured that the simulated precipitation remained reasonable. By investigating high intensity events of the recent time period, the observed precipitation measured in the present network exceeded the precipitation measured at the historical network in more than 91% of the cases, and the mean of the exceedances was 61%. Thus, the limit of the distribution defined in Equation (5) was rather conservative.

The spatial variability of the precipitation amounts was described with the help of a variogram. As the number of historical observations was usually low, the variogram was estimated from the

mean normalized variograms $\gamma_M(h)$ of the daily amounts for month $M$ of the present times. This was rescaled with the variance of the distribution function $F_{t_j}(z)$.

$$\gamma_{t_j}(h) = \text{Var}(Z(x, t_j))\gamma_{M(t_j)}(h) \tag{6}$$

The spatial distribution for each day was generated using the method described in Hörning et al. [30]. The objective function for the generation was the mean squared error of the simulated discharge. The method was applied for each separate day in a random sequence. It was repeated to obtain a set of precipitation realizations. It is important to note that the constraint discussed before on having a distribution for the entire grid is a much looser one than constraining the catchment distribution, which in this case is very uncertain as there are only two observation stations in the catchment.

One could consider simulating the precipitation in 3D i.e., taking time as the third dimension. Here we did not do so because for daily precipitation in this region, the temporal correlations (Pearson) are around 0.2 which are mostly made up of days with no precipitation.

## 3. Results

### 3.1. Calibration and Validation Performance

Using DE, for the present day calibration and validation period NSEs were 0.88 and 0.85 respectively. The calibrated model parameter values are shown in Table 1. Using ROPE for a total of 441 parameter vectors, calibration NSEs were all around 0.86 while they lied between 0.80 to 0.84 for the validation. The parameter vectors differed from each other quite strongly. Figure 2 shows the variability of the model parameters.

**Table 1.** HBV model parameter values for present and historical calibrations using DE. Parameter values of suspicious magnitude are in **bold**.

| Parameter | Units | Minimum | Maximum | 1991–2000 | 1882 |
|---|---|---|---|---|---|
| $TT$ | °C | −1 | 1 | 0.21 | **0.84** |
| $CM_{TE}$ | mm/day/°C | 0 | 4 | 2.94 | **0** |
| $CM_{PR}$ | mm/day/°C/mm | 0 | 2 | 0.14 | 0.31 |
| $FC$ | mm | 1 | 700 | 385 | **16** |
| $\beta$ | - | 0 | 7 | 2.44 | **1.62** |
| $PWP$ | mm | 1 | 700 | 323 | **16** |
| $UT$ | mm | 0 | 100 | 7 | 26 |
| $K_{uu}$ | 1/day | 0 | 0.7 | 0.27 | **0.70** |
| $K_{ul}$ | 1/day | 0 | 0.6 | 0.14 | 0.35 |
| $K_d$ | 1/day | 0 | 0.7 | 0.30 | 0.65 |
| $K_{ll}$ | 1/day | 0 | 0.3 | 0.08 | 0.06 |

Model performance for one large snow melt event in calibration and validation periods each are shown in Figures 3 and 4. The model reproduced both the events well. Thus, we believed that it was suitable for modelling the historical event.

For the historical flood event, the model was run for the whole year of 1882, but results were evaluated for the months of November and December only because those were the months with simulated precipitation. Using interpolated precipitation as an input, the calculated discharges differed from the observed ones. Figure 5 shows the observed and the modelled discharges.

The NSE for the model was 0.55 (last two months only) using the parameter vector from DE based on present day calibration. Using ROPE, the NSEs remained in the range of 0.39 and 0.61. Figure 6 shows the corresponding discharge series for the year of 1882. Please note that all these vectors performed well for the present time period. They did not match the observed series which, in all cases, was out of the range of the simulated series.

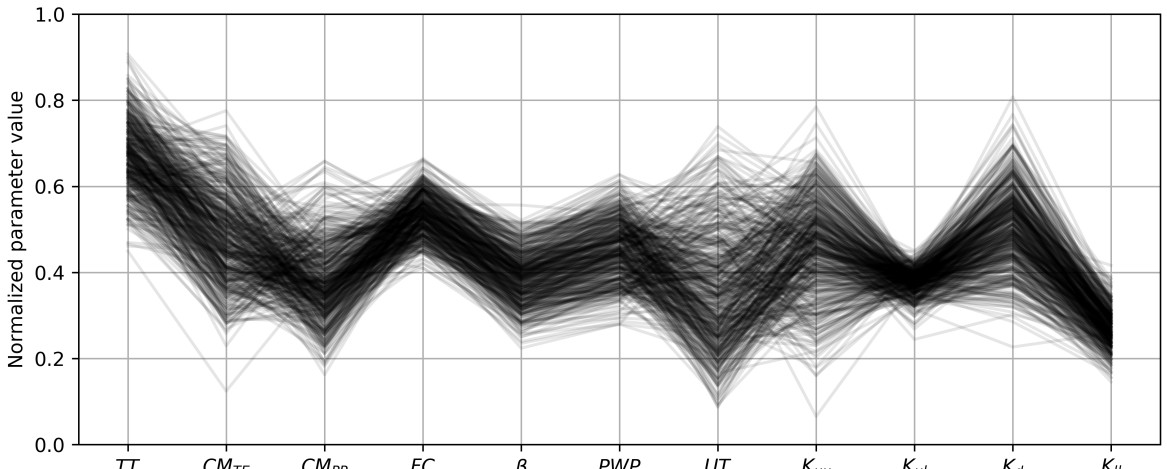

**Figure 2.** Well performing model parameters (N = 441) for the present day calibration period using ROPE.

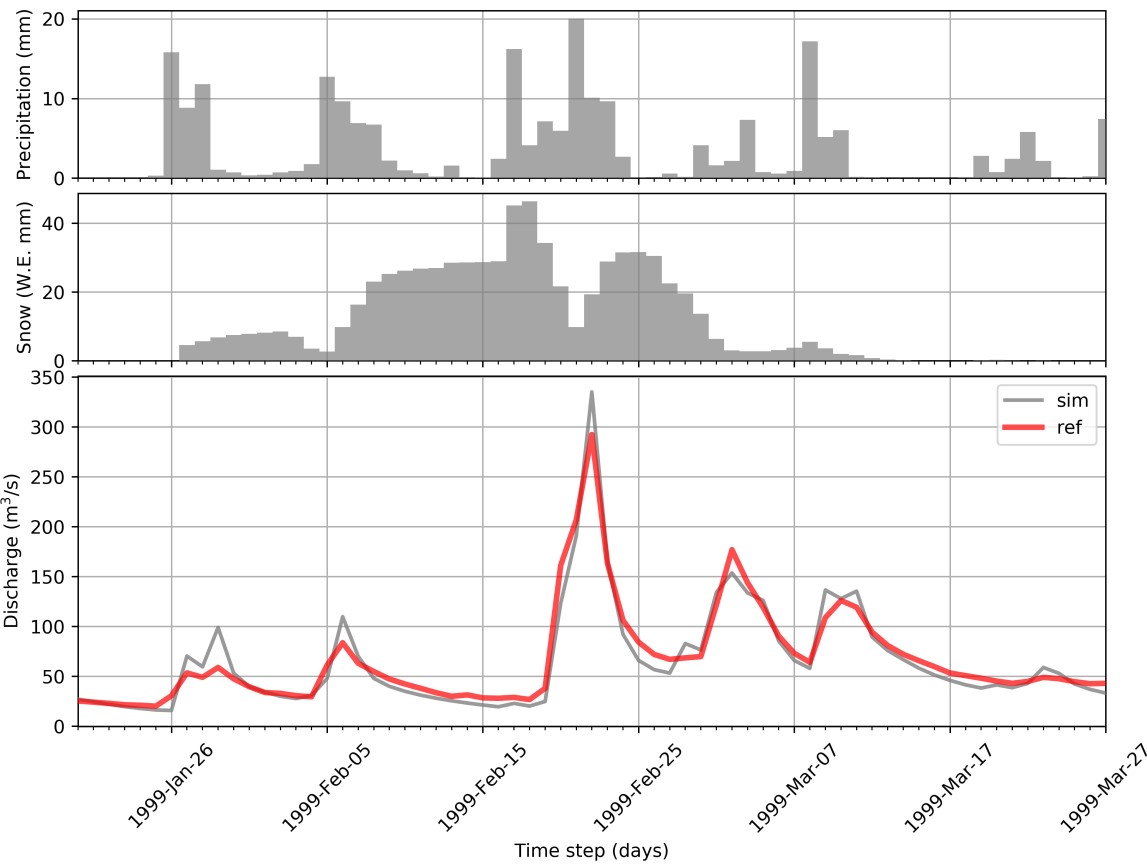

**Figure 3.** A calibration period large snow melt event using DE.

To make sure that the data was the problem and not the model, calibration was performed for the year of 1882 with the first three months as the warm up period using DE and ROPE using the historical data. This resulted in an NSE of 0.75 with quite a reasonable hydrograph using DE (Figure 7).

ROPE resulted in NSEs of around 0.72. But the resulting model parameter vectors had very questionable properties (**bold** in Table 1). $CM_{TE}$ converged to zero for DE while $CM_{PR}$ converged to 0.35. $FC$, $PWP$ and $\beta$ converged to 16, 16, and 1.6 respectively. These values show considerable departure from values that are common for this area, based on our experience. For example, a value

of 16 for *PWP* means that the amount of model evapotranspiration equals the PET if soil moisture is more than 16 mm. This almost always results in maximum evapotranspiration, a scenario that is highly unlikely.

Using these parameters for the present day time period resulted in NSEs of −0.70 and −0.47 for calibration and validation periods respectively using DE. This can also be attributed to the fact that calibrations on short time series result in an over fit but here the problem is much more severe i.e., having no snow melt due to temperature rise is physically not possible. Using ROPE, the parameter vectors seemed more reasonable (Figure 8), except for *TT* which tended to 1 (1 was the upper limit). This behaviour has the same effect as setting the $CM_{TE}$ parameter to zero. For the present calibration period, the NSEs lied between −0.76 and −0.50 for calibration and between −0.51 and −0.26 for validation. To verify that ROPE and DE were producing same type of parameter vectors, $CM_{TE}$ was fixed to 3 with the rest of the settings being the same in DE. The optimization was performed again. This time *TT* went to the upper limits i.e., 1, same as ROPE.

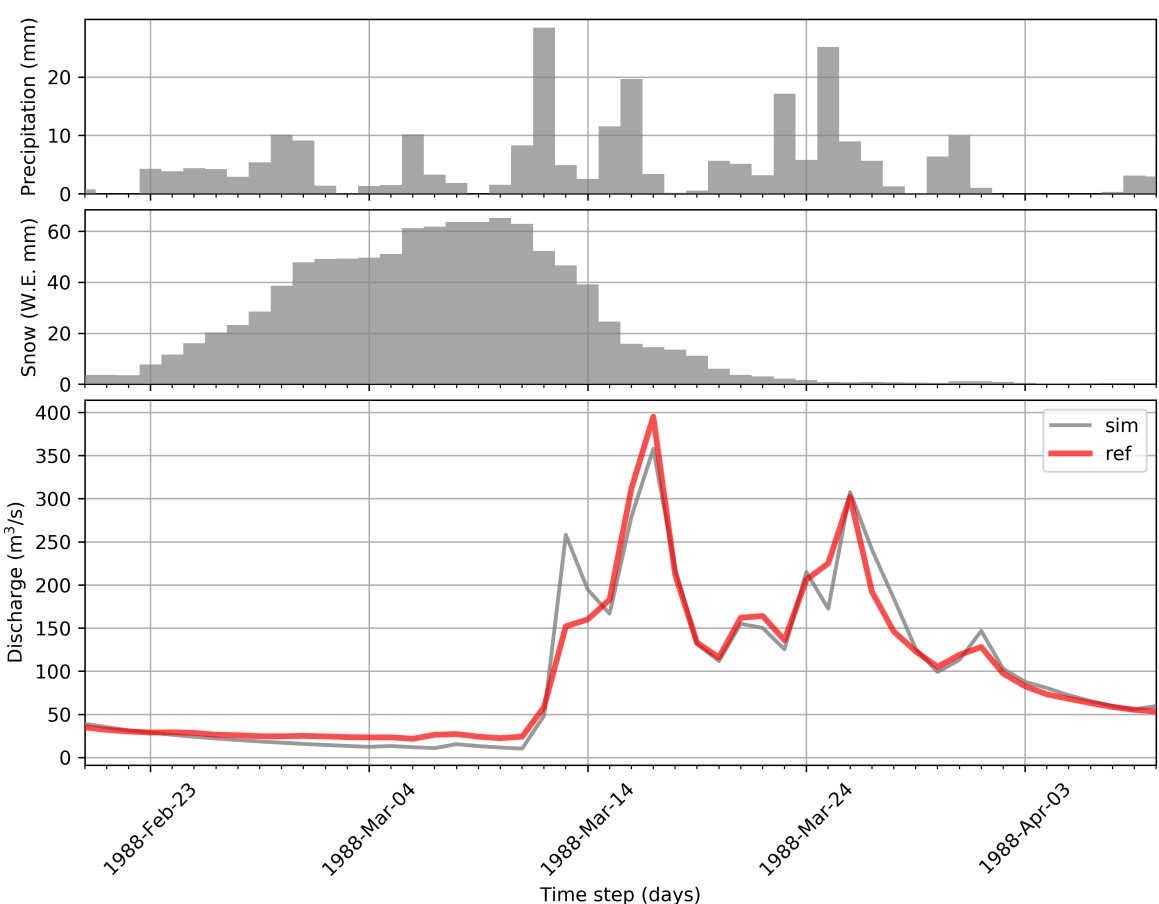

**Figure 4.** A validation period large snow melt event using DE.

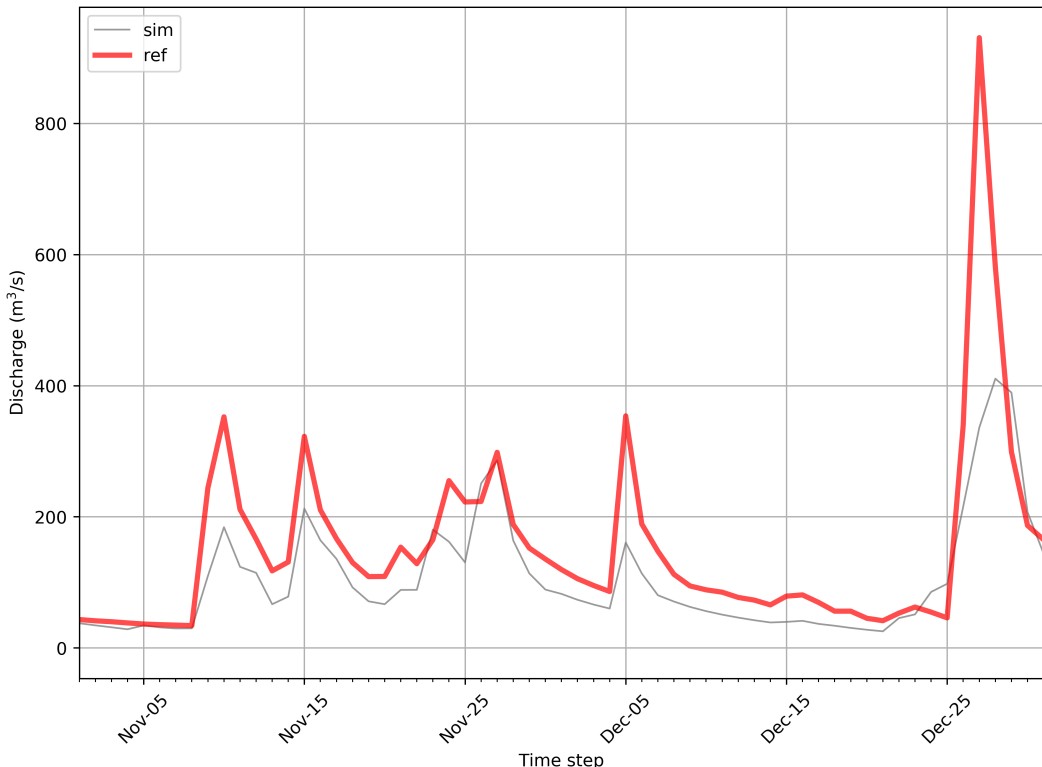

**Figure 5.** Observed (red) and calculated (gray) discharges for November and December 1882 using parameters from the present calibration period.

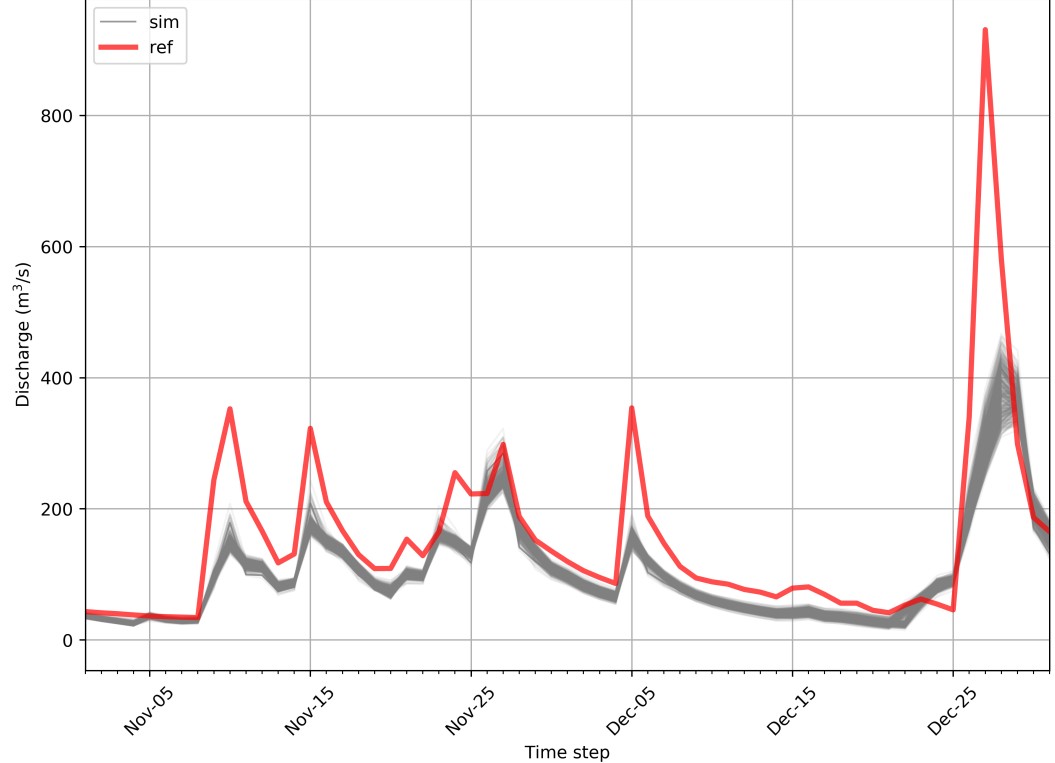

**Figure 6.** Observed (red) and calculated (gray) discharges for November and December 1882 using the interpolated precipitation and 441 different parameter sets.

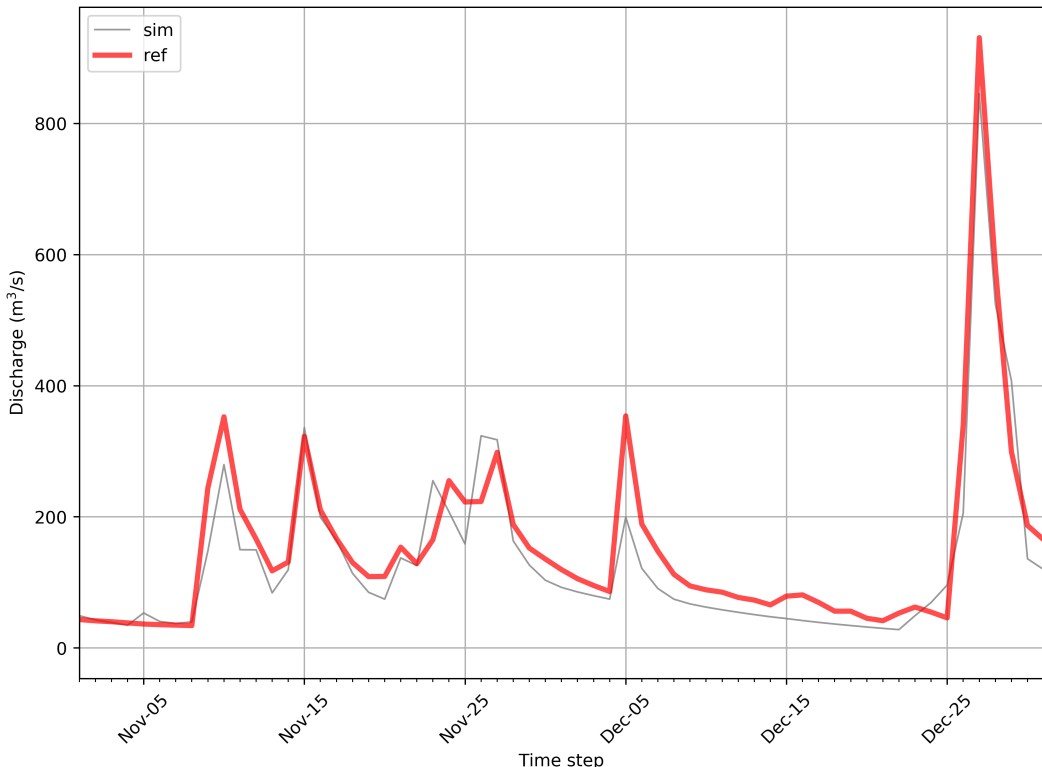

**Figure 7.** Observed (red) and (gray) calculated discharges for November and December 1882 using parameters from the historical period.

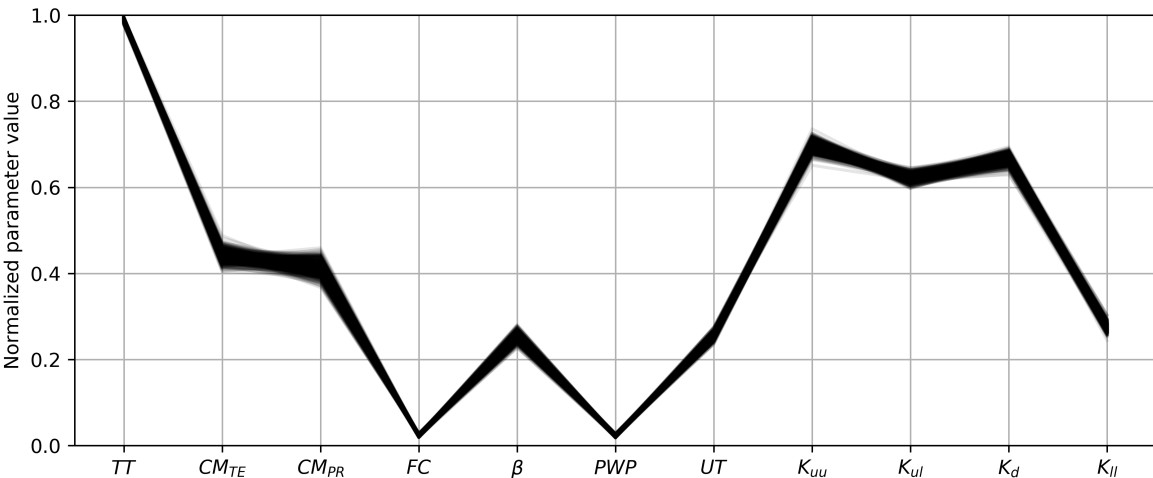

**Figure 8.** Well performing model parameters (N = 441) for the year of 1882 using ROPE.

*3.2. Inverted Precipitation Performance*

For the inversion, 20 days with high precipitation in the months of November and December were selected. In total, 100 different inverted spatially distributed rainfall series were generated. For the precipitation field simulation a 200 km × 200 km domain was selected so that the boundary conditions had negligible influence. Within this area, 20 locations with precipitation measurements were available for conditioning only (cf. Figure 1).

Figure 9 shows the distribution of daily precipitation amounts for 24 December 1882. The non-parametric fit of the distribution was very good. Please note that the upper tail of the distribution was modified with the maximum being 10% higher than the observed maximum according

to Equation (5). The modified non-parametric fit allowed unobserved precipitation amounts to occur in the simulated fields.

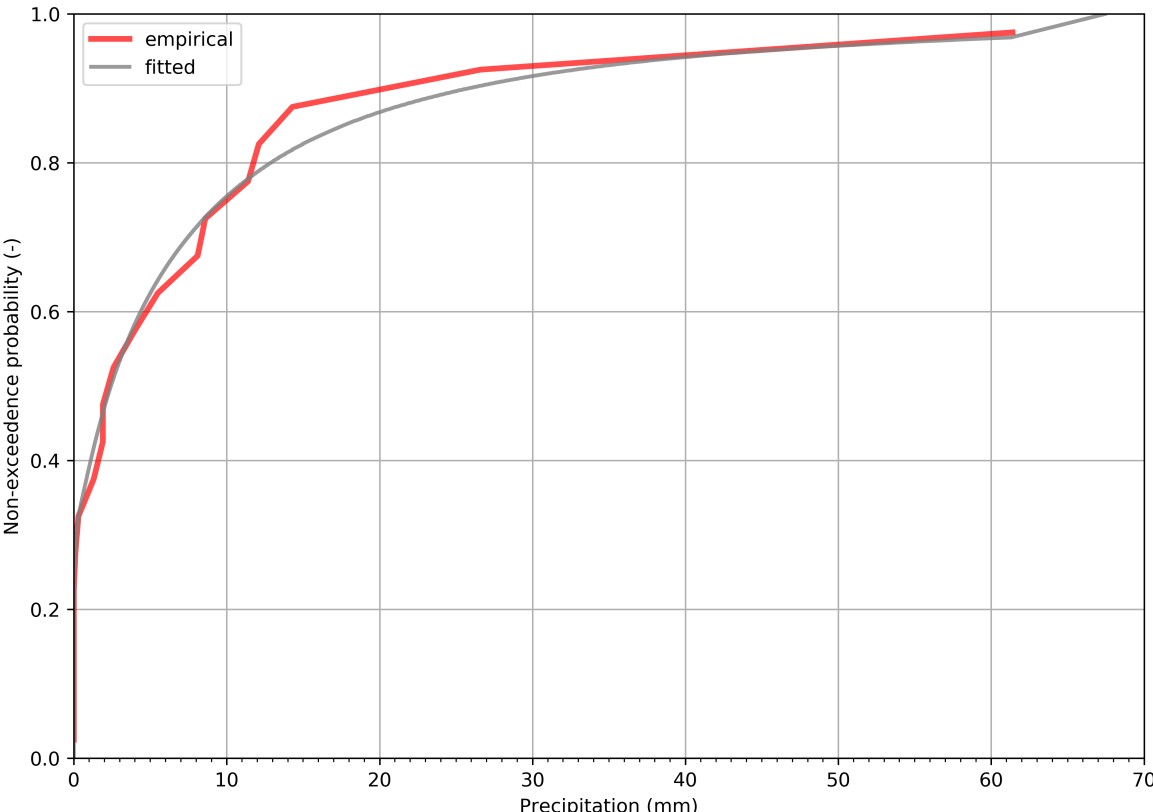

**Figure 9.** Empirical and fitted non-parametric distributions for the 24 December 1882 event.

　　　Due to the small number of observation points the uncertainty of the fields was high i.e., the individual realizations were very different from each other. Figure 10 shows four realizations for 24 December 1882. All 100 realizations were different, the upper two represent the most dissimilar realizations, while the lower shows the most similar pair. Similarly, Figure 11 shows four realizations for 25 December 1882.

　　　Subsequently, the hydrological model was run using all realizations. Figure 12 shows the corresponding discharge series. They all fitted the observed discharge well despite their differences in spatial structure. The resulting NSEs were all between 0.97 and 0.98, showing that the fit was very good.

　　　In the case when all 441 ROPE parameter vectors were used in combination with the 100 precipitation realizations the obtained series fitted the observed ones much better, even though the model was calibrated using DE. Figure 13 shows the corresponding discharge series. All NSEs stayed between 0.82 and 0.98. This showed that the uncertainty in precipitation can explain bad model performance better than the uncertainty in model parameters, at least in our case. Still, for the many parameters' case, the peak showed quiet a wide band (750–1350 $m^3 s^{-1}$), demonstrating model parameter uncertainty.

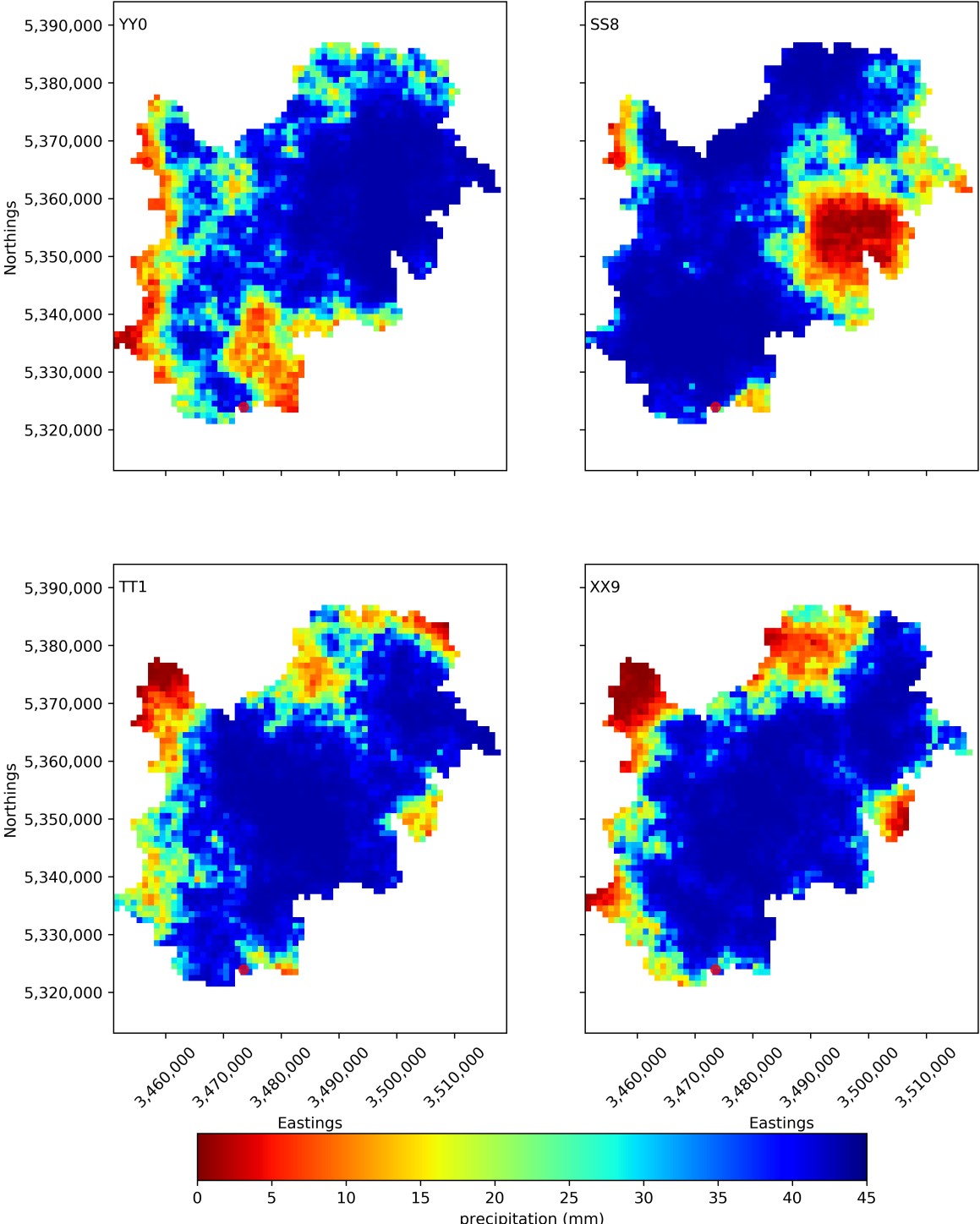

**Figure 10.** Gridded inverted precipitation fields for 24 December 1882. The upper two panels show the most dissimilar simulated pair, the lower the most similar pair.

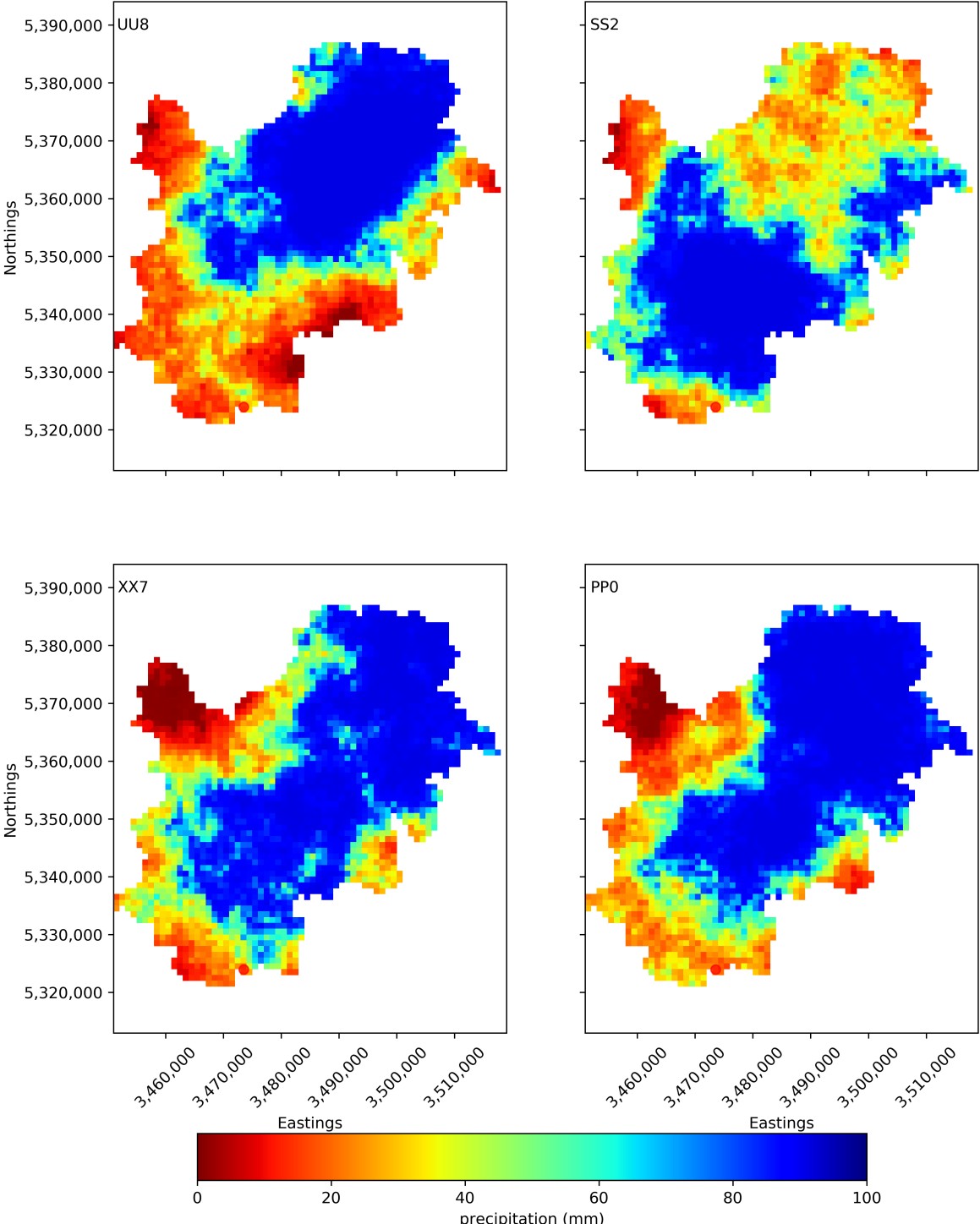

**Figure 11.** Gridded simulated precipitation fields for 25 December 1882. The upper two panels show the most dissimilar simulated pair, the lower the most similar pair.

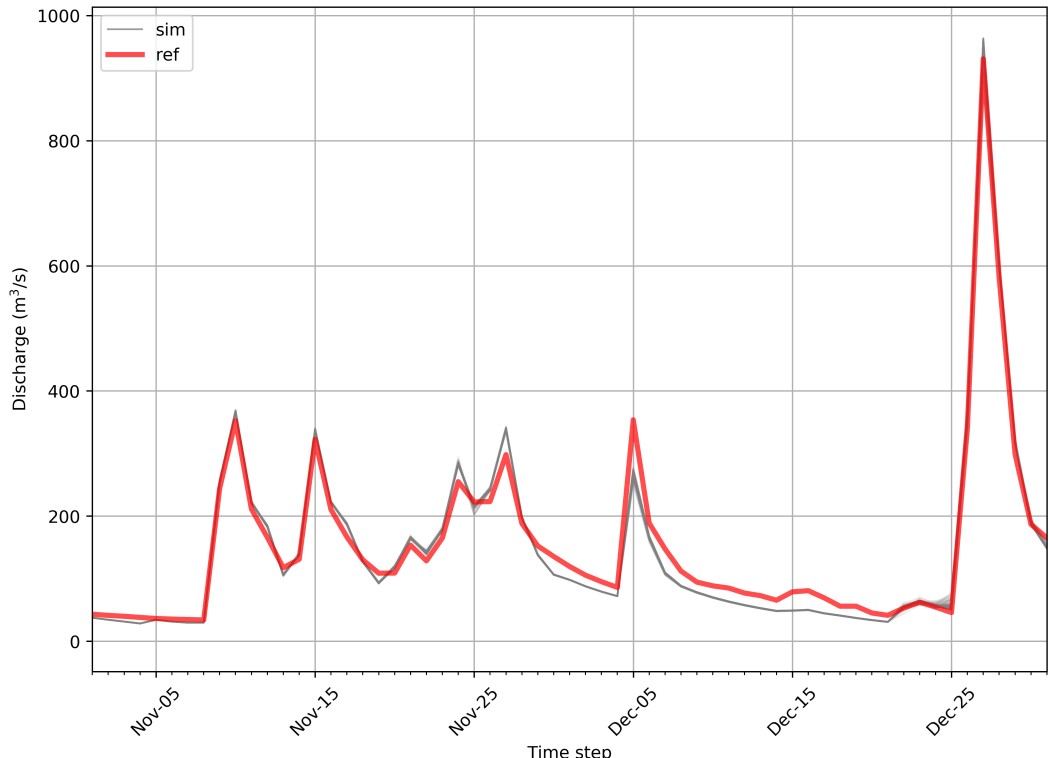

**Figure 12.** Observed (red) and calculated (gray) discharges for the November and December 1882 using the 100 simulated realizations.

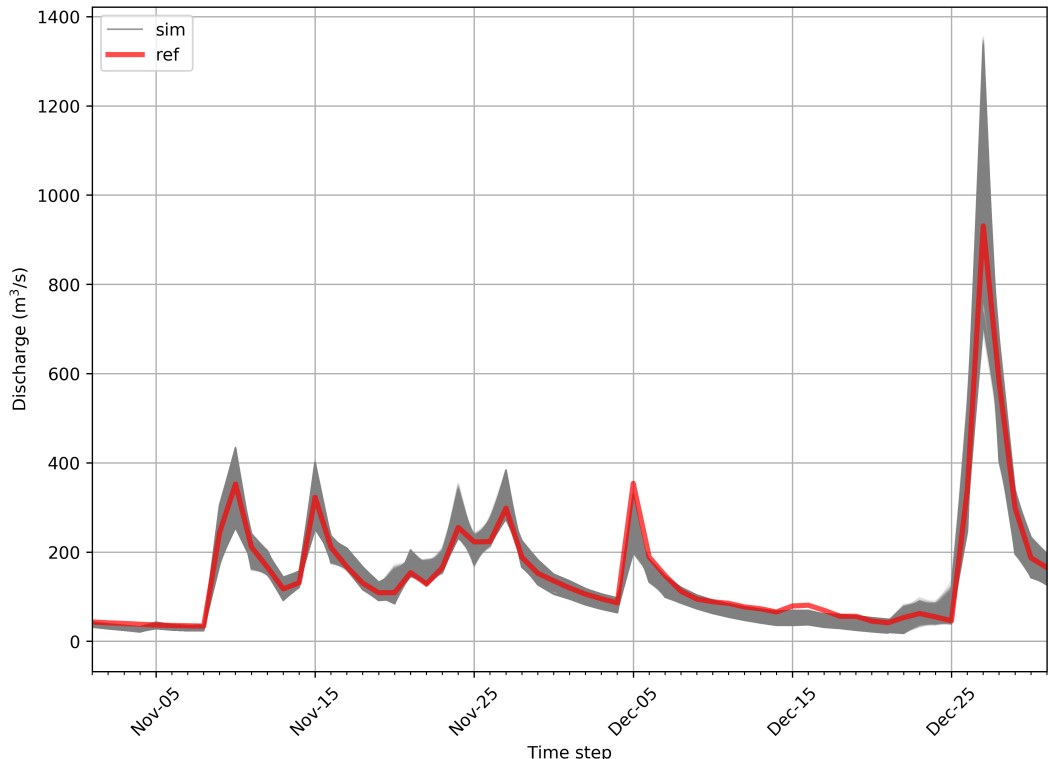

**Figure 13.** Observed (red) and calculated (gray) discharges for November and December 1882 using the 100 precipitation realizations and 441 different parameter sets.

### 3.3. Inverted Precipitation Plausibility

The simulated rainfall fields used as input into the hydrological models ensured a good estimate of the observed discharge. The fields were all constrained to the observations in and around the catchment. This itself does not imply that the precipitation fields have a reasonable and meteorologically plausible spatial distribution. In order to check the plausibility of the spatial patterns, observations of 55 present years (1961–2015) were considered. For this time period, 36 to 57 observation stations measuring daily rainfall in and around the catchment were selected.

For each simulated field, precipitation amounts generated at the locations of the recent stations were extracted. The correlations (Pearson and Spearman) between the daily precipitation amounts measured in the recent past and those obtained from all simulated fields were calculated i.e., each simulation time step per realization compared against all days in the present time period. If for a simulated realization a high correlation was found then the observed pattern was deemed similar. Overall, Pearson correlations of selected analogues varied between 0.79 and 0.92 while Spearmans between 0.40 and 0.78.

For all realizations, reasonable analogues in recent observations could be found. Figure 14 shows the precipitation field interpolated from observations on 11 May 2009 and the precipitation field interpolated using values of a specific simulation (VV7) at the observation locations used for the 24 December 1882 event. The two maps are very similar in their spatial structure.

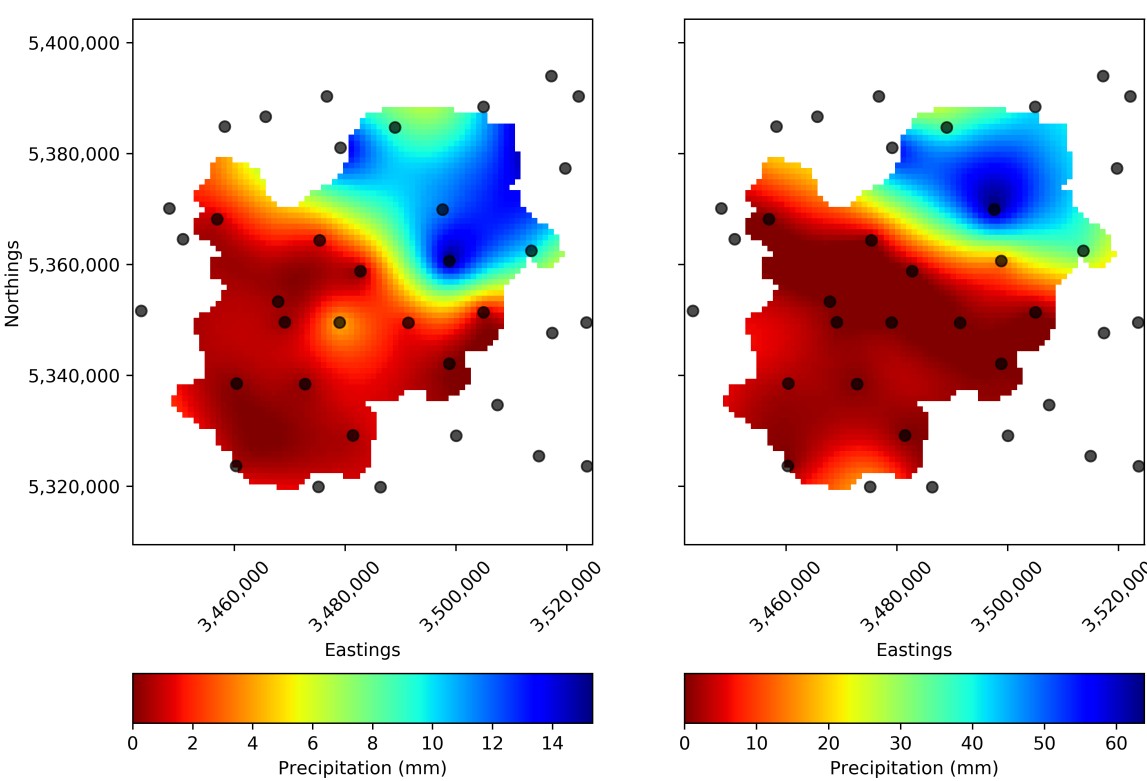

**Figure 14.** Observed precipitation (**left**) interpolated from observed data on 11 May 2009 and interpolated (**right**) using simulated values (VV7) at the observed locations on 24 December 1882.

It should be noted that the maximum for the entire field on the 24 December was set to around 70 mm of precipitation globally (Figure 9) but the maximum simulated precipitation settled around 43 mm locally (Figure 10). Same happened for the 25 December. This goes to show that to match the peak, simply increasing the magnitude of precipitation beyond observation bounds was not the solution.

### 3.4. Snowmelt Contribution to the Peak

Subsequently, the processes leading to the flood event were investigated using the hydrological model. Snowmelt contributed to the flood significantly. The amount of snowmelt related to the event varied between 14 and 22% (Figure 15).

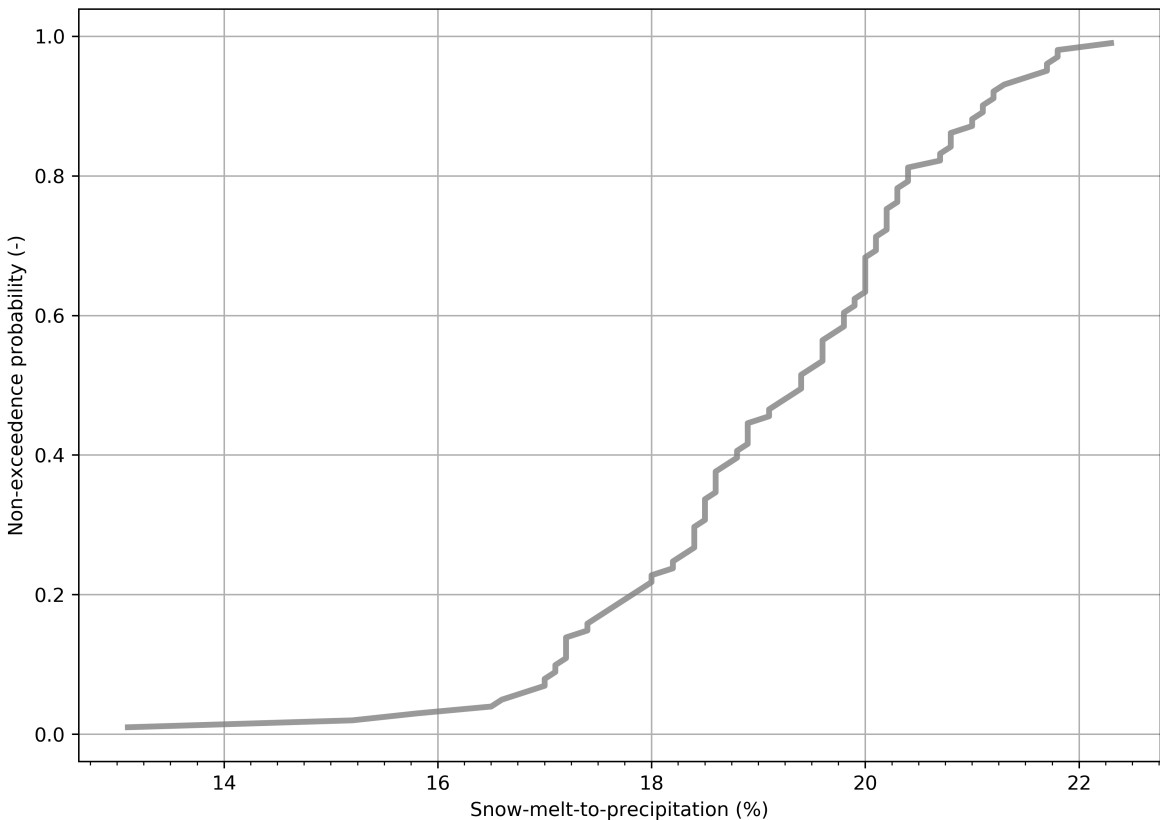

**Figure 15.** Contribution of snowmelt to the flood peak for 100 realizations.

Figure 16 shows two contrasting sequences of modelled snow water equivalents between the 22 and 26 December. The corresponding precipitation amounts for the same period are shown in Figure 17. One can see that the patterns show some similarity, but there is quite a difference between the spatial distributions.

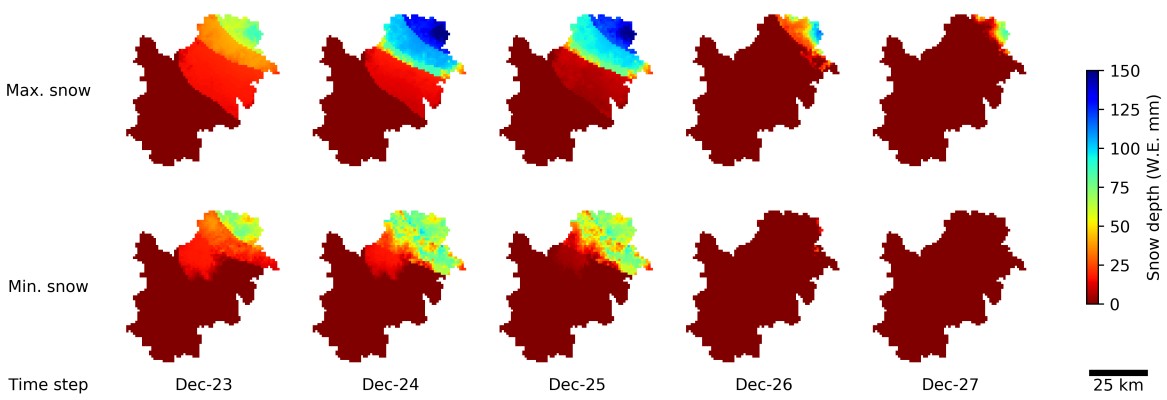

**Figure 16.** Available snow as snow water equivalent for two contrasting realizations.

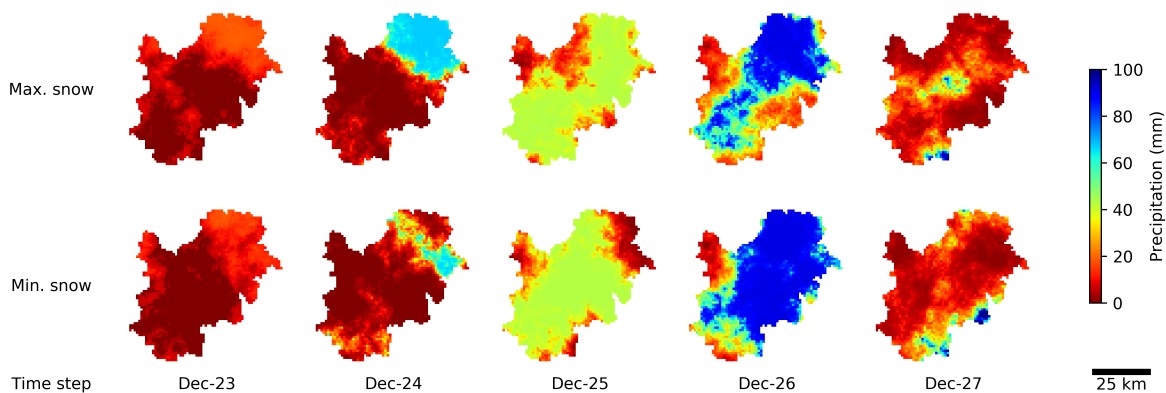

**Figure 17.** Precipitation on days before the flood for two contrasting realizations.

## 4. Discussion and Conclusions

Hydrological model output uncertainty is usually associated with the parameters of the model [31]. In this paper, we investigated the effects of precipitation uncertainty in reproducing a historical flood event in data sparse conditions. Using interpolated historical data lead to a bad model performance. This was mainly due to data sparsity as only two gauges in the catchment were unlikely to capture high precipitation amounts. On the other hand, we simulated reasonable precipitation realizations which respected both the observations and their spatial variability and at the same time enabled a very good reproduction of the observed discharge. A large number of such realizations was generated out of the infinitely many possible, showing the high uncertainty caused by the small number of precipitation observations.

For the historical event, a systematic problem with interpolated precipitation, apart from the magnitude, was the timing of the peaks. All parameter vectors resulted in peaks that were one step too late. This was also the case in Seidel et al. [6], where the modelled peak was very close to the reconstructed one but arrived a step later. One reason for peak under estimation could be the sheer magnitude of the event. The reconstructed-rescaled historical peak at Kirchentellinsfurt was about 930 $m^3s^{-1}$ while the maximum recorded peak in the present day data is 538 $m^3s^{-1}$. Such a big difference can be an indication that processes of such magnitude taking place in those times do not repeat any more, as pointed out by Fischer and Schumann [8] for a nearby region in Germany. Yet another reason could be that the spatial interpolations using very few control points result in smooth fields, where the chances for a point measurement to capture the peak value are almost none, resulting in an overall underestimation.

Even though Seidel et al. [6] modelled the hydrometeorological event quite well, here the focus was to demonstrate that in data sparse conditions model parameter uncertainty is not the only cause of concern and for the same parameter vectors, a rather infinite amount of reasonable inputs can be used to simulate the outputs.

Hydrological model uncertainty is usually associated with the parameters of the model Beven and Binley [31], but we believe that in such situations precipitation uncertainty can explain model errors well, while parameter uncertainty cannot. Thus, modelling in data sparse regions is very uncertain. Model parameters should be estimated by taking precipitation uncertainty into account.

As a side note, parameters of conceptual hydrological models depend on the available observations as well. Point precipitation is interpolated for the catchment and used as input for the model. This input itself is very uncertain and in some cases even biased. In Bárdossy and Das [32] it was shown that models calibrated on dense networks may perform bad if used with data from sparse networks, and models calibrated on sparse networks may perform bad if used with data from dense networks.

**Author Contributions:** Conceptualization, methodology, visualization, A.B., J.S. and F.A.; software, A.B. and F.A.; writing—original draft preparation, A.B., J.S. and F.A.; supervision, A.B.; project administration, A.B. and J.S.; funding acquisition, A.B., J.S. and F.A. All authors have read and agreed to the published version of the manuscript.

**Funding:** This research was conducted within the research group FOR 2416 "Space-Time Dynamics of Extreme Flood" founded by the German Research Foundation (Deutsche Forschungsgemeinschaft- DFG; BA 1150/22-2).

**Acknowledgments:** The authors acknowledge the funding and financial support of the research group FOR 2416 "Space-Time Dynamics of Extreme Floods (SPATE)" by the German Research Foundation (DFG). Most of the software used here was based on Python, its libraries and accompanying software (Numpy, Pandas, SciPy, Matplotlib, Cython, Pathos and H5py, Eclipse IDE, PyDev, EGit and MikTex). We are very thankful to the developers/contributors that eased the effort required for modelling and presenting the results of this paper by providing the basic building blocks. We would also like to thank the German Weather Service (DWD) and the Landesanstalt für Umwelt Baden-Württemberg (LUBW) for making their data available for free to the public. Finally, we also appreciate the comments and suggestions of the three anonymous reviewers that helped to improve this text and the editor who oversaw the rather fast review process.

**Conflicts of Interest:** The authors declare no conflict of interest.

## Abbreviations

The following abbreviations are used in this manuscript:

| | |
|---|---|
| DE | Differential Evolution |
| HBV | Hydrologiska Byråns Vattenbalansavdelning model |
| IDW | Inverse distance weighting |
| NSE | Nash-Sutcliffe efficiency |
| OK | Ordinary Kriging |
| PET | Potential evapotranspiration |
| ROPE | Robust Parameter Estimation |
| W.E. | Water Equivalent |

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
