# Peer review of "Hydrological Modelling in Data Sparse Environment: Inverse Modelling of a Historical Flood Event"

_water, doi:10.3390/w12113242_

Round 1
Reviewer 1 Report
- The study addressed the effects of precipitation uncertainty on modelling historical floods in data sparse conditions. It is not clear how the land use changes were accounted for the simulation.
- Check the equations (4) and (5). What is # in equation (4)? Why is the area used in equation (4)? Why the resulting distribution is 1 in equation (5)?
- In addition to NSE, the coefficient of persistence can be used for assessing k-step lead time forecasting model.
Kitanidis PK, Bras RL (1980) Real-time forecasting with a conceptual hydrologic model, 2, applications and results. Water Resour Res 16(6):1034–1044
- Section 3.2 needs to be described in Section 2 because the section 3.2 explains the characteristics of hydrometeorological conditions not the results.
- It is not clear whether or not the simulated result in Figure 7 was obtained by historical precipitation. If then, what is the difference of peak discharge error between the simulations with historical precipitation and generated precipitation.
- The simulated discharge responses in Figures 12 and 13 were in good agreement with the observed response despite the spatial variability of 100 precipitation realizations applied for the simulation is quite different. It is not clear of the reason for it. Is it due to the discharge responses conditioned for the precipitation generation? Is there any physical explanation?
- There is a need to revise the incorrect sentences and words in the manuscript. For example,
Line 68: where a hydrograph for the year 1882 is available Seidel et al. [6], does not exist any more
Line 86: OK was not chosen this time as there are very few stations
Line 70 : 2.320 km2 whereas
Line 71: of 1.900
Line 72: a factor of 1,22
Line 289: precipitation, lead to bad a model performance
Reviewer 2 Report
This study analyses a historical flood event in Germany using HBV model and the inverse modelling method. Results clearly indicated that the optimization of the model parameters performed well to capture historic flood event. I think the paper is well written and recommend to publish it as its current form. Just minor suggestions are below: - P 2 L91–93: It would be helpful if some more sentences about why HBV model was chosen for their analysis. - Figure 3 and others: Generally, observations are shown in gray and simulations are shown in colored lines.Author Response
Please see the attachment.

Reviewer 3 Report
The submitted paper deals with model output uncertainty in data sparse region, precisely it considers how input and model parameter uncertainties affect the output. The paper is a continuation of the previous research conducted by one of co-authors. It is certainly original and contains results that will contribute to the advance in current knowledge. The research was carried out reliably, using the available data and appropriate tools with high standards of presentation. Finally, they reach right/proper conclusions. I think that it will be interesting for many readers of the Water Journal. That's why I recommend it for publication as is.
Author Response
Made no comments.
Round 2
Reviewer 1 Report
The authors responded properly to my previous comments.